# Rapid Customization and Manipulation Mechanism of Micro-Droplet Chip for 3D Cell Culture

**DOI:** 10.3390/mi13122050

**Published:** 2022-11-23

**Authors:** Haiqiang Liu, Chen Yang, Bangbing Wang

**Affiliations:** 1School of Mechanical Engineering, Hangzhou Dianzi University, Hangzhou 310018, China; 2School of Earth Sciences, Zhejiang University, Hangzhou 310058, China

**Keywords:** micro-droplet chip, manipulation, cell culture, dispersed phase, continuous phase

## Abstract

A full PDMS micro-droplet chip for 3D cell culture was prepared by using SLA light-curing 3D printing technology. This technology can quickly customize various chips required for experiments, saving time and capital costs for experiments. Moreover, an injection molding method was used to prepare the full PDMS chip, and the convex mold was prepared by light-curing 3D printing technology. Compared with the traditional preparation process of micro-droplet chips, the use of 3D printing technology to prepare micro-droplet chips can save manufacturing and time costs. The different ratios of PDMS substrate and cover sheet and the material for making the convex mold can improve the bonding strength and power of the micro-droplet chip. Use the prepared micro-droplet chip to carry out micro-droplet forming and manipulation experiments. Aimed to the performance of the full PDMS micro-droplet chip in biological culture was verified by using a solution such as chondrocyte suspension, and the control of the micro-droplet was achieved by controlling the flow rate of the dispersed phase and continuous phase. Experimental verification shows that the designed chip can meet the requirements of experiments, and it can be observed that the micro-droplets of sodium alginate and the calcium chloride solution are cross-linked into microspheres with three-dimensional (3D) structures. These microspheres are fixed on a biological scaffold made of calcium silicate and polyvinyl alcohol. Subsequently, the state of the cells after different time cultures was observed, and it was observed that the chondrocytes grew well in the microsphere droplets. The proposed method has fine control over the microenvironment and accurate droplet size manipulation provided by fluid flow compared to existing studies.

## 1. Introduction

Microfluidic cell culture integrates knowledge from biology, biochemistry, engineering, and physics to develop devices and techniques for culturing, maintaining, analyzing, and experimenting with cells at the micro-scale. A key component of a microfluidic cell culture chip is being able to mimic the cell microenvironment which includes soluble factors that regulate cell structure, function, behavior, and growth. For example: co-cultivation and interaction between cells, construction and simulation of in vitro cell microenvironment, single-cell manipulation, bacterial culture and analysis, and chip organs, etc. [1,2,3]. Microfluidic technology mainly forms droplets through the insolubility of two or more liquids. The droplets formed are generally in the range of several micrometers to hundreds of micrometers [4]. Although there have been many cell analyses performed on microfluidic chips, various cell analyses are based on cell culture [5]. The two-dimensional (2D) cell culture using the microfluidic chip method is similar to the traditional cell culture in a Petri dish or flask, in which cells are attached to the substrate. Among them, the supply of nutrients, waste liquid produced by metabolism, and drug stimulation are all provided by the culture medium [6,7]. Before performing a two-dimensional (2D) cell culture, the surface of the flow channel will be treated to facilitate the growth of cells. The usual treatment is to change the hydrophilicity of the flow channel surface. Most of the treatment methods are collagen solution or Soak in polylysine solution [8]. Then, plant the cells in the chip, and then let it stand. After the cells are stably attached to the bottom of the flow channel of the chip, add cell culture medium [9]. Although 2D cell culture has the above-mentioned advantages, the real biological environment still cannot be restored in this way [10]. The contact is a kind of direct contact, which is bound to cause unpredictable damage to the cells, and this damage is caused by the large shear stress generated by the 2D flow [11]. 

To sum up the multiple defects of 2D cell culture, the microenvironment of the cells in these defects is quite different from the real cell growth environment in vivo, so two-dimensional cell culture cannot truly and accurately reflect the true reaction of the cells in the body and will result in experimental deviations, so that the reliability and authenticity of the experimental results cannot meet the predetermined requirements. Three-dimensional (3D) cell culture just solves the problem of authenticity simulated in 2D cell culture and can reflect the normal anatomical and physiological characteristics of cells. The most common way to establish a three-dimensional cell culture environment on a microfluidic chip is to grow cells in a variety of natural or synthetic materials that mimic the extracellular matrix, such as Matrigel [12], collagen [13], water, gel [14,15,16], and so on. The cell growth process requires biological scaffolds to provide attachment points for them, and the above-mentioned extracellular matrix simulation materials can provide such biological scaffolds. Wu et al. [17] developed a 3D culture microfluidic chip based on simulating the diffusion process between blood vessels and tissues. In addition to materials other than hydrogels, a variety of gels have been applied to the cell culture process [18,19,20]. Toh added the micro-column structure to the microfluidic chip, successfully realized 3D cell culture, and maintained the normal detoxification function of liver cells [21].

In summary of the various shortcomings and shortcomings of two-dimensional (2D) cell culture, this article is based on droplet microfluidics to study the rapid customization method and manipulation mechanism of micro-droplet chips for three-dimensional (3D) cell culture [22]. Because the traditional chip manufacturing process is too complicated, the cost of processing equipment and processing time is limited, so this article uses 3D printing technology to print a convex mold for injection molding to make micro-droplet chips; and after comparing the T-channel method [23], the flow focusing method [24,25] and the coaxial flow method [26] and the other three micro-droplet forming methods, by comparing the advantages and disadvantages of each [27,28]. Finally, the flow-focusing method was chosen as the droplet-forming method of the microfluidic chip.

## 2. Materials and Methods

### 2.1. Materials

Microchannels are fabricated by casting the mixture of polydimethylsiloxane (PDMS) elastomer and a curing agent (Silpot 184 W/C, Dow Corning Toray Co., Ltd., Tokyo, Japan) against a 3D-printed mold. The mold was fabricated with a photocurable resin (High Temp Resin, Formlabs, Somerville, MA, USA), using a commercial stereolithography machine.

The preparation process of the micro-droplet chip is shown in Figure 1. The preparation of the micro-droplet chip is divided into three steps: ① SLA light-cured printing chip convex mold, ② different ratio of PDMS preparation, and ③ full PDMS microfluidic chip molding. Among them, A-E is the printing process of the chip convex mold, F-G is the PDMS preparation process of different proportions, and H-L is the full PDMS microfluidic chip molding process.

### 2.2. Fabrication

Using additive manufacturing, the positive mold required to prepare the full PDMS micro-droplet chip is printed out. The manufacturing steps are as follows:(1)Modeling of chip convex mold

Design the flow channel structure of the micro-droplet chip and save it in STL format (Figure 2).

(2)Parameters setting and printing

Formlabs company provides a dedicated data processing software environment. The software provides a simple function of automatically generating support and layout, and also provides the function of customizing the support and layout. When making a model layout and adding support, the following requirements need to be met: The principle of 45° angle;The principle of least support;The principle of single-angle start. Combining the above three principles, taking into account the subsequent requirements of the chip microchannel pouring Bonding re-quires surface roughness and transparency. It is not advisable to add too many sup-ports to the chip support plane. Therefore, we choose a cylindrical support with a den-sity of 100% and a printing inclination of 60° for chip printing (Figure 3).

(3)Post-printing processing

Take off the forming platform, insert it into the bracket, and then use a special shovel to remove the formed die model of the chip. Place the formed chip convex mold model in the cleaning box containing isopropanol solution on the left and soak for 10 min and shake it from time to time to clean the residual resin on the surface of the model, and then place it on the right and clean with 98% ethanol solution. Rinse in the box for 5–10 min to remove the isopropanol liquid on the chip. Place the model in a dust-free environment for 5~10 min to completely volatilize the ethanol on the surface of the chip, and also make the surface of the chip not fully cured for secondary curing.

After drying, use a special tool to remove the model. Use 2500 mesh sandpaper to polish the surface with support residues to obtain the preliminary finished product of the chip.

(4)Finished product on the surface of the substrate convex mold.

Use a microscope to observe the surface of the printed substrate convex mold, as shown in Figure 4, from (a,b) there are obvious surface impurities, and (c,d) it can be seen that the shrinking part is in line with the design requirements. Therefore, it is necessary to purge the surface of the micro-droplet chip in order to meet the requirements of the experiment.

### 2.3. Casting of Full PDMS Micro-Droplet Chip

(1)Preparation of the mold

Substrate mold production: use cardboard to make a rectangular paper cofferdam with a length and width slightly larger than the length and width of the substrate convex mold by 3 mm, and then place the substrate convex mold in the cofferdam to form a PDMS replication mold. The cover slip mold uses a Petri dish with a diameter of 75 mm as the mold.

(2)The ratio and mixing of PDMS and curing agent

In order to improve the chip packaging strength, it can be achieved by changing the ratio of the matrix/curing agent in the PDMS cover sheet and the substrate. The main reason is: when the ratio of the matrix/curing agent in the PDMS cover sheet and the substrate is different, the adhesion interface layer produces molecular diffusion, thus forming a strong adhesion transition layer. Therefore, in this paper, the substrate is prepared according to the ratio of PDMS and curing agent of 10:1, and the cover sheet is prepared according to the ratio of PDMS and curing agent of 15:1. 

(3)Degassing to remove bubbles

The prepared substrate PDMS and cover sheet PDMS were placed in a transparent vacuum chamber for 30 min. The pressure in the vacuum chamber was maintained at −1 bar through a vacuum pump. As shown in Figure 5, it can be seen from Figure 5b that a lot of bubbles emerge in a negative pressure environment. These bubbles are generated during the preparation and stirring process.

(4)Pouring PDMS on the mold

Spray an appropriate amount of release agent on the substrate mold and cover sheet mold in advance to separate the molded PDMS substrate and cover sheet; then, respectively, the substrate PDMS and cover sheet PDMS close to the edge of the mold and slowly pour into the mold, avoid new bubbles during the pouring process.

(5)PDMS cutting and punching.

Cut the two strips of PDMS into a 60 mm × 40 mm rectangle, and then punch holes in the corresponding position of the cover sheet PDMS with a diameter of 1.5 mm, as shown in Figure 6.

(6)PDMS baking and stripping mold

Put the prepared PDMS in a constant-temperature drying oven for 40 min. The temperature of the drying oven is constant. The removed substrate PDMS and cover piece PDMS are air-cooled to room temperature. And then the two pieces of PDMS are slowly peeled out of the mold. When the temperature in the drying box is set to 80 °C, the effect of PDMS is shown in Figure 7a–f.

The surface-forming effect of the runner is better without too obvious defects; but there are obvious cracks on the surface of the chip in contact with the mold. As shown in Figure 7g–i, obvious cracks appeared in the mold during the drying process. It is estimated that the temperature is too high, so appropriately reduce the temperature during drying. 

After several adjustments, it is found that the PDMS substrate has no previous cracks at 75 °C, as shown in Figure 8. Place the poured PDMS in a constant-temperature drying oven for 40 min. The temperature of the drying oven is constant at 75 °C. Air-cool the removed substrate PDMS and cover PDMS to room temperature, and then carefully and slowly peel off the two pieces of PDMS from the mold.

(7)Bonding of PDMS chip

The bonding method of the PDMS chip can be roughly divided into the natural sealing method [29] or the sealing method after surface treatment [30]. The first natural sealing method is to use the affinity between the two pieces of PDMS, and without adding adhesives, it is possible to simply bond the PDMS substrate with the flow channel and the PDMS cover sheet to complete low-strength bonding. However, this kind of bonding must be unbreakable. When the outside is filled with liquid or cleaned, it may cause the separation of the two pieces of PDMS, thereby destroying the tightness of the chip. The second method of sealing after surface treatment solves the problem of bonding strength, but the instruments used for oxygen plasma treatment are relatively expensive equipment, which greatly increases the cost of the experiment [31]. The microvalves and micropumps are made with different ratios of PDMS and curing agent, and higher bonding strength is obtained. Therefore, the ratio of the PDMS substrate in this article is 10:1, and the ratio of the PDMS cover sheet is 15:1, just to obtain higher bonding strength. Before bonding, each component needs to be cleaned with isopropanol to remove all dust and particles on the surface. Put them in the Petri dish and put them in a constant temperature drying oven for 1 h to compare the bonding strength. The temperature in the drying oven is constant at 75 °C, and a test chip is designed to conduct a tightness test. The test results are shown in Figure 9. The combined strength meets the requirements of the experiment, and there is no side leakage after the dying agent is introduced.

## 3. Experiments and Results

The mixture of pure water and sodium alginate solution is used as fluid flow 1 of laminar two-phase flow and level set coupling, and mineral oil is used as fluid flow 2 of laminar two-phase flow and level set coupling. The basic parameters of the two fluids are shown in Table 1.

According to the three sets of data of the disperse phase flow rate X1 as 100 μL/h, 150 μL/h, 200 μL/h, set the continuous phase flow rate X2 with the difference of 10 arithmetic sequence to increase slowly, and observe the drop size situation, the experimental results are shown in Figure 10.

The measured data shows that when the flow rate of the dispersed phase remains unchanged, as the flow rate of the continuous phase gradually increases, the droplets formed gradually decrease, as shown in Figure 11, where L/W is the micro ratio of the length of the droplet to the width of the flow channel. This trend is generally consistent with the simulation results using COMSOL Multiphysics finite element software.

There are many ways to establish a geometric model in COMSOL Multiphysics software. The first is to establish the geometric model of the flow channel structure in the built-in drawing module of COMSOL Multiphysics software, but this method is difficult for drawing complex shapes; the second is to draw the geometric model in AutoCAD and other drawing software and import it. The total length of the channel structure of the micro-droplet chip is 20.51 mm and the total width is 9.82 mm.

According to the continuous phase X2 flow rate of 100 μL/h, 150 μL/h, and 200 μL/h three sets of data, set the dispersed phase flow rate X1 to slowly increase the arithmetic sequence with a difference of 10 for introduction, and observe the droplet size change In this case, the experimental results are shown in Figure 12. The measured data shows that when the flow rate of the continuous phase remains unchanged, as the flow rate of the dispersed phase gradually increases, the droplets formed gradually increase (Figure 13).

Based on comparing the data and relation from droplet size and the phase transition in Figure 11 and Figure 13, we can draw the conclusion that when the flow rate of the dispersed phase remains constant, as the flow rate of the continuous phase gradually increases, the droplets formed gradually decrease. When the flow rate of the continuous phase remains constant as the flow rate of the dispersed phase gradually increases, the formed droplets also gradually increase.

The sodium alginate micro-droplets and 3% calcium chloride solution were subjected to an ion cross-linking reaction. The core of the cross-linking reaction is that the calcium ions in the calcium chloride solution replace the sodium ions on the sodium alginate G unit to form an “egg-box” structure. This structure can greatly improve the mechanical strength of the gel, so that the gel meets the requirements of mechanical properties. When the micro-droplet undergoes a cross-linking reaction, the micro-droplet becomes a micro-spherical droplet with a 3D structure. At this time, the droplet has certain mechanical properties and mechanical properties. The micro-droplets after cross-linking are shown in Figure 14a–c.

Calcium silicate powder and polyvinyl alcohol (PVA) solution are mixed to prepare a printing material, and the three-dimensional scaffold structure is printed out by a 3D-bioplotter printer and sintered to obtain a biological scaffold for supporting microspheres, as shown in Figure 14d,e. The micro-droplet of sodium alginate-containing cells is formed by the micro-droplet chip and dropped onto the rectangular biological scaffold soaked in calcium chloride solution. Due to the cross-linking reaction between the sodium alginate micro-droplets and the calcium chloride solution, the micro-droplets become microspheres and are fixed on the biological scaffold, as shown in Figure 14f–i. 

After the cells were cultured for 0 h, 24 h, and 48 h, the Petri dish containing the microsphere droplets with 3D morphology was taken out of the incubator. Place the Petri dish under the YDF-880 fluorescence microscope for microscopic observation, and take pictures of the cell culture through the image acquisition function of the microscope. The culture of chondrocytes on the chip is shown in Figure 15. The picture shows the growth of chondrocyte cells. It can be seen from Figure 15 that the cells are growing well, and the cells have begun to divide and pass down. This verifies that the chip customized in this article can perform 3D cell culture well and meet the experimental requirements for 3D cell culture.

## 4. Conclusions

Microfluidic chips are now widely used in many fields such as gene mutation detection, genotyping, DNA sequencing, cell culture, cell sorting, and cell state research. These works mainly focused on the design and rapid customization of micro-droplet chips for three-dimensional (3D) cell culture, and apply SLA light-curing 3D printing technology to the preparation process of micro-droplet chips, making the preparation of micro-droplet chips faster and more convenient and more economical. The performance of the full PDMS micro-droplet chip in biological culture was verified by using a solution such as chondrocyte suspension, and the control of the micro-droplet was achieved by controlling the flow rate of the dispersed phase and continuous phase. Experimental verification shows that the designed chip can meet the requirements of the experiments, and it can be observed that the micro-droplets of sodium alginate and the calcium chloride solution are cross-linked into microspheres with three-dimensional (3D) structures. These microspheres are fixed on a biological scaffold made of calcium silicate and polyvinyl alcohol. Subsequently, the state of the cells after 0 h, 24 h, and 48 h culture was observed, and it was observed that the chondrocytes grew well in the microsphere droplets. All the results presented in this work verified that the chip design and manufacturing process used are feasible. It shows prosperous applications in the field of life science due to its nontoxicity and biocompatibility.

This microfluidic chip is easy to manipulate and quantify for customizing. The proposed method has fine control over the microenvironment and accurate droplet size manipulation provided by fluid flow compared to existing studies.

## Figures and Tables

**Figure 1 micromachines-13-02050-f001:**
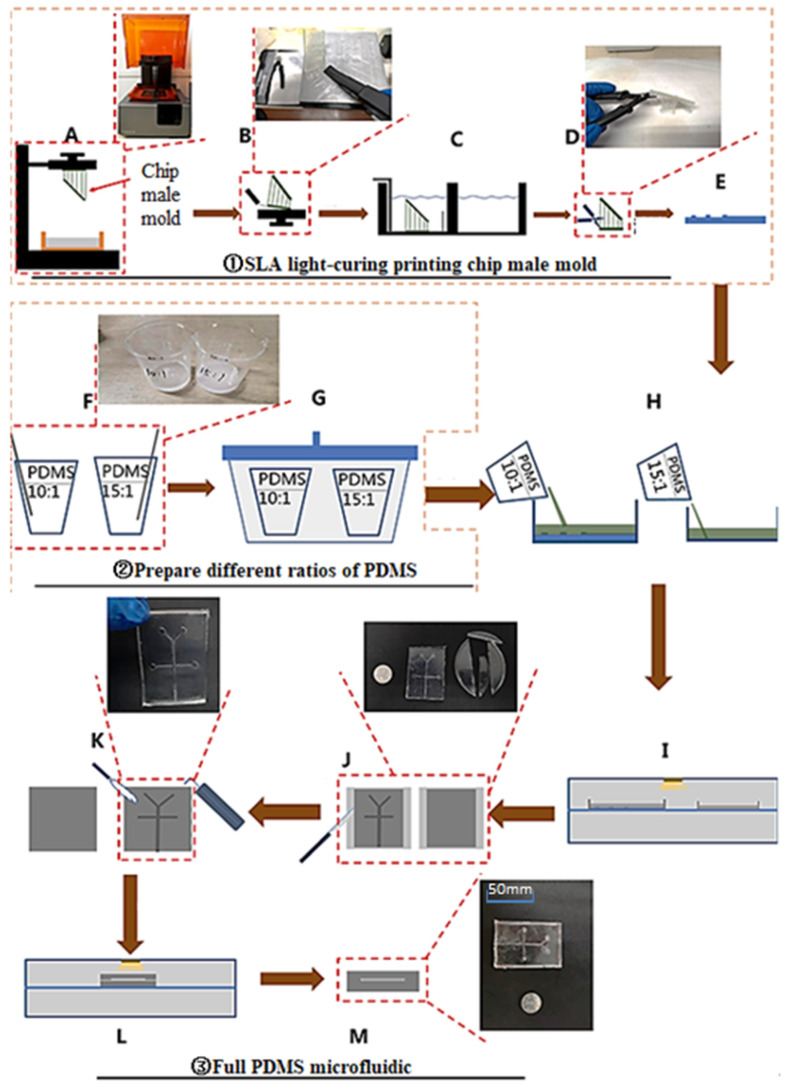
Schematic diagram of the preparation process of the micro-droplet chip. (**A**–**E**) Printing process of the chip convex mold; (**F**–**G**) The PDMS preparation process of different proportion; (**H**–**M**) Full PDMS microfluidic chip molding process.

**Figure 2 micromachines-13-02050-f002:**
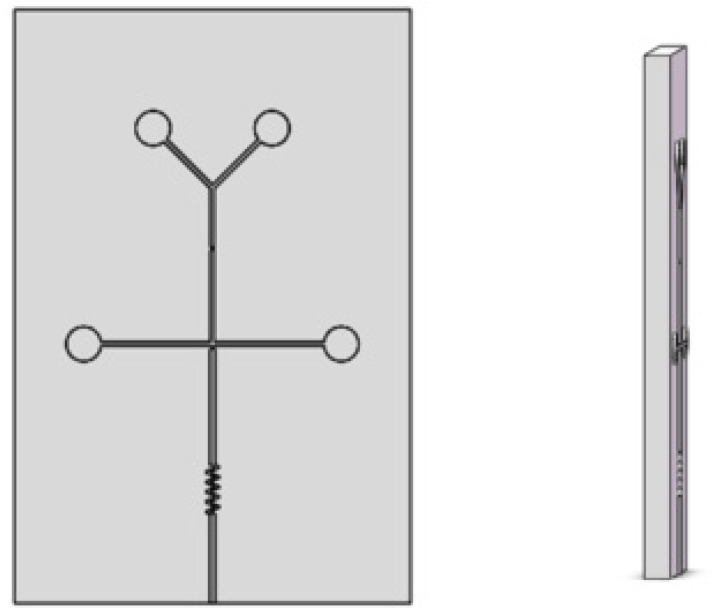
Design three-dimensional drawing of mold.

**Figure 3 micromachines-13-02050-f003:**
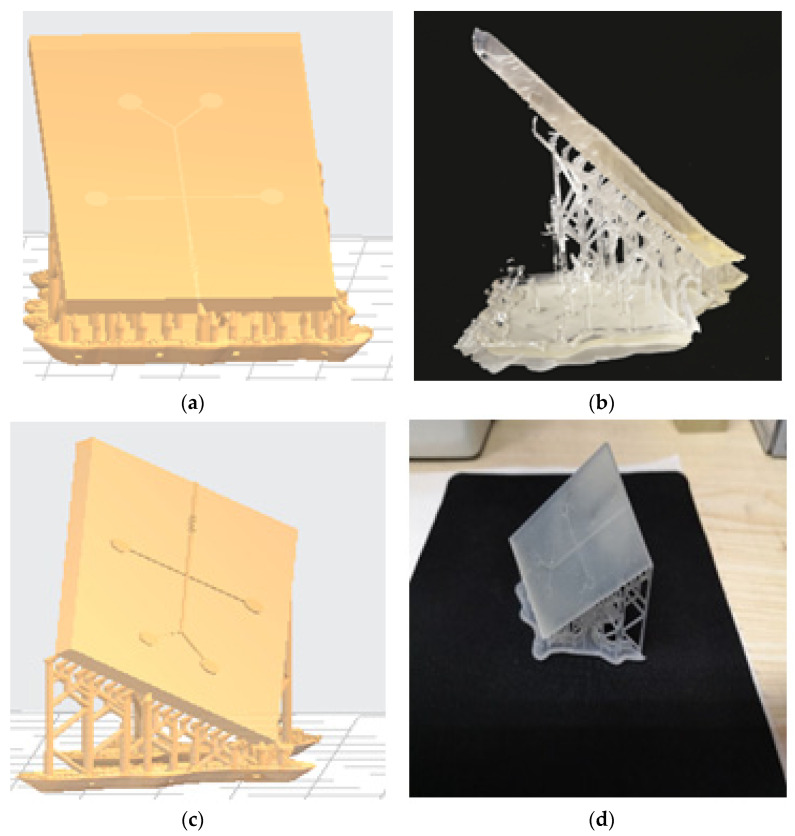
(**a**) Non-single-angle printing three-dimensional model; (**b**) non-single-angle printing effect diagram; (**c**) single-angle printing three-dimensional model; (**d**) single-angle printing effect.

**Figure 4 micromachines-13-02050-f004:**
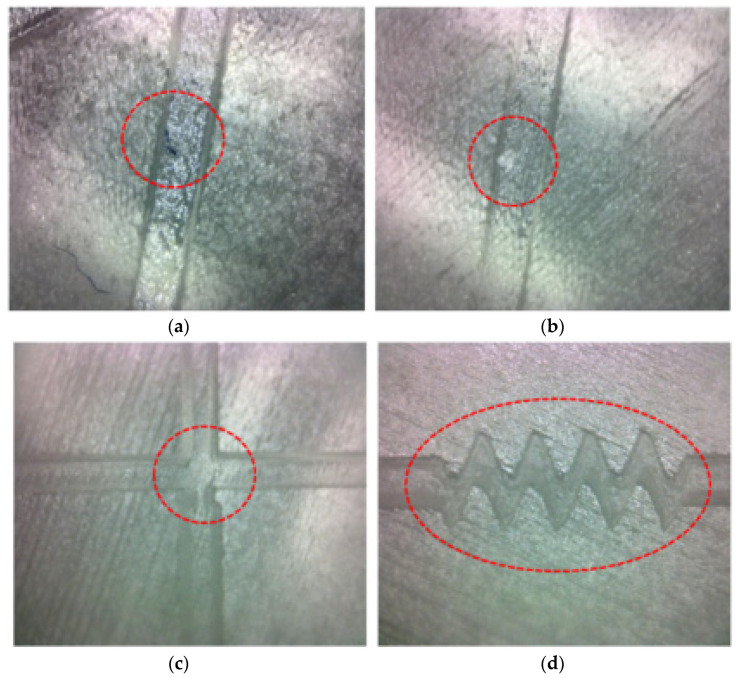
(**a**,**b**) There are obvious surface impurities problems. (**c**,**d**) It can be seen that the forming at the shrinkage part meets the design requirements.

**Figure 5 micromachines-13-02050-f005:**
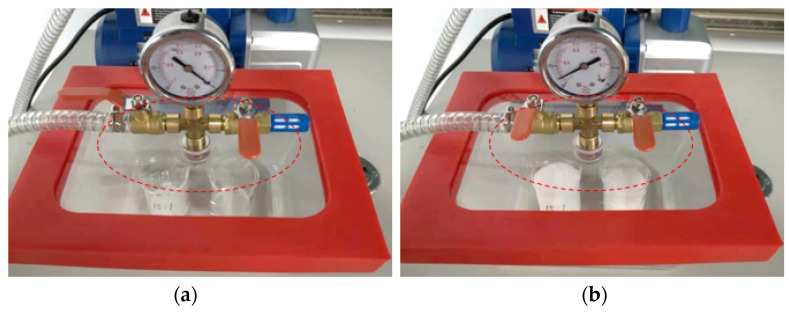
(**a**) Status of PDMS before pressurization; (**b**) status of after pressurization.

**Figure 6 micromachines-13-02050-f006:**
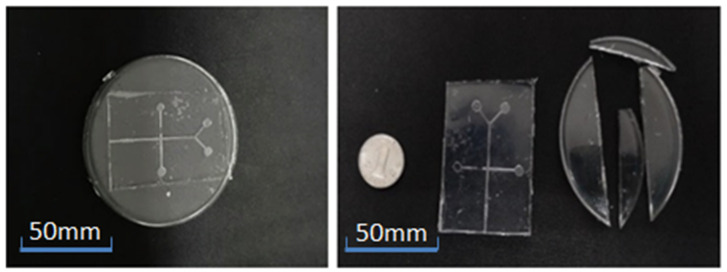
Chip before cutting and after cutting.

**Figure 7 micromachines-13-02050-f007:**
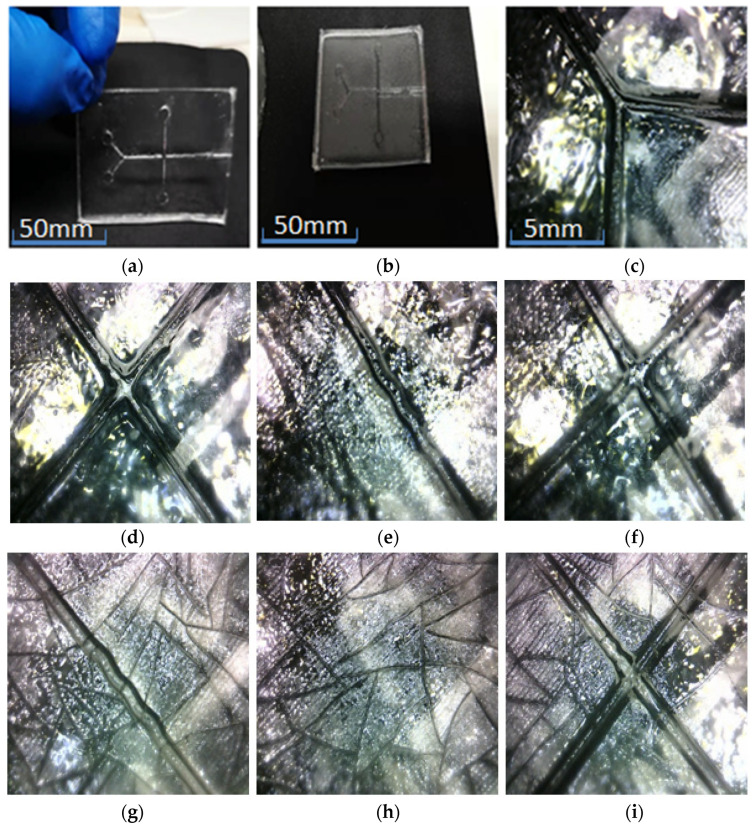
Molding state after drying at 80 °C. The surface-forming effect of the runner shown in the diagrams (**a**–**f**) is good, without too obvious defects; but there are obvious cracks on the surface of the chip in contact with the mold. The convex mold of the substrate shown in (**g**–**i**) has obvious cracks during the drying process.

**Figure 8 micromachines-13-02050-f008:**
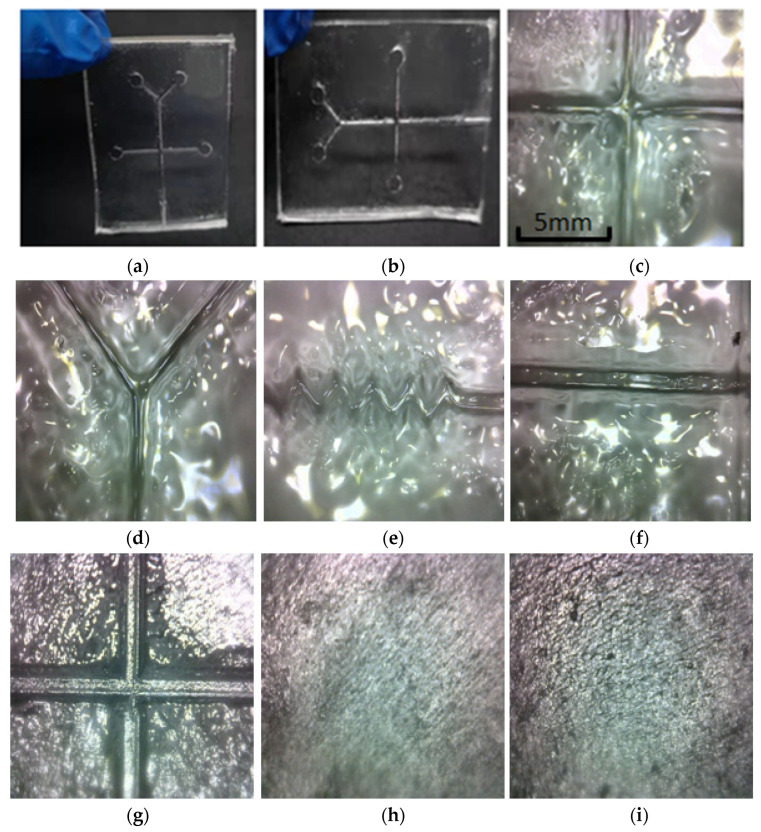
Molding state after drying at 75 °C. Figures (**a**–**f**) show that the surface forming effect of the runner is better without too obvious defects; (**g**–**i**) shows that the substrate mold does not appear obvious cracks during the drying process.

**Figure 9 micromachines-13-02050-f009:**
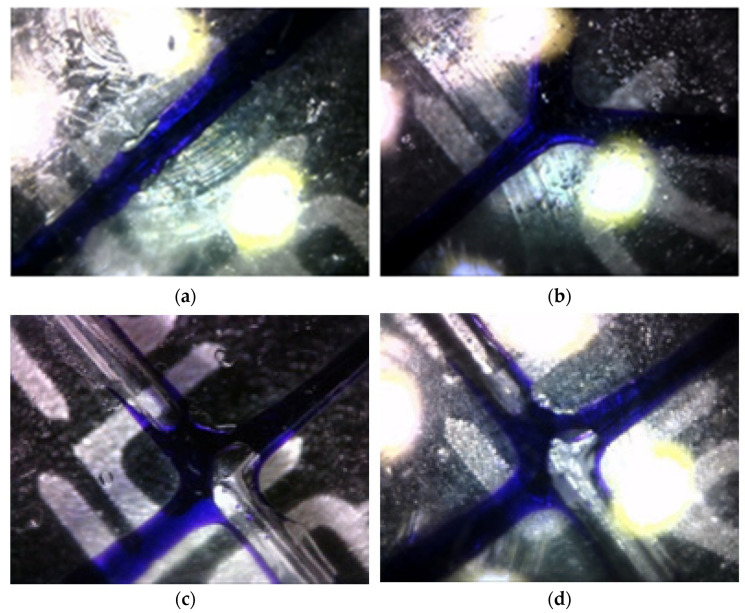
Experimental diagram of the tightness of chip. (**a**,**b**) drying for one hour; (**c**,**d**) heating in an oven at 75 °C.

**Figure 10 micromachines-13-02050-f010:**
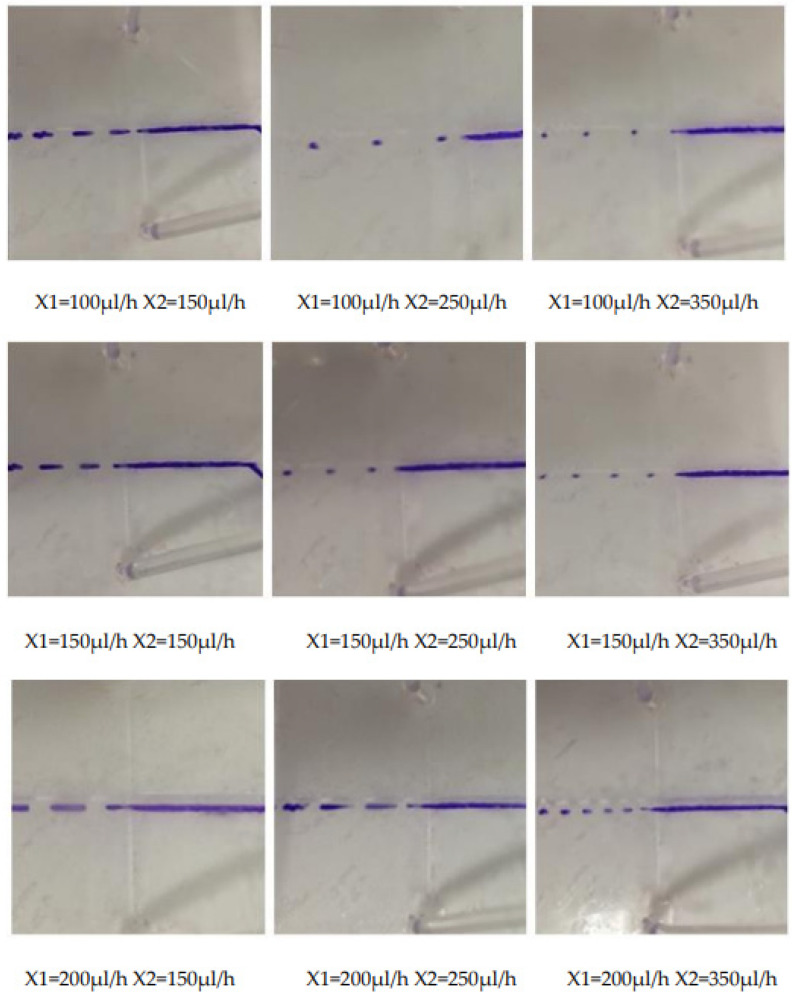
The experimental results of flow rate of the dispersed phase being constant and the flow rate of the continuous phase changing.

**Figure 11 micromachines-13-02050-f011:**
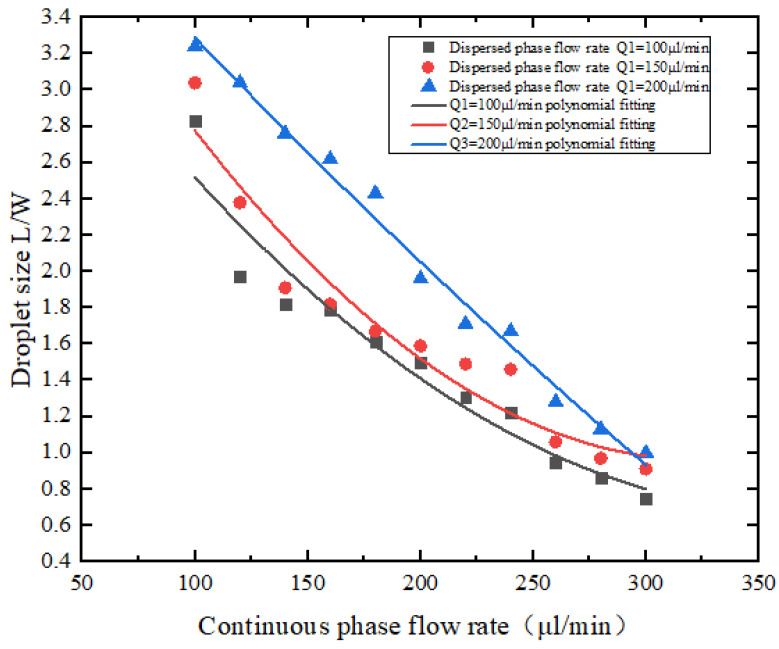
The relationship between continuous phase flow rate and droplet size.

**Figure 12 micromachines-13-02050-f012:**
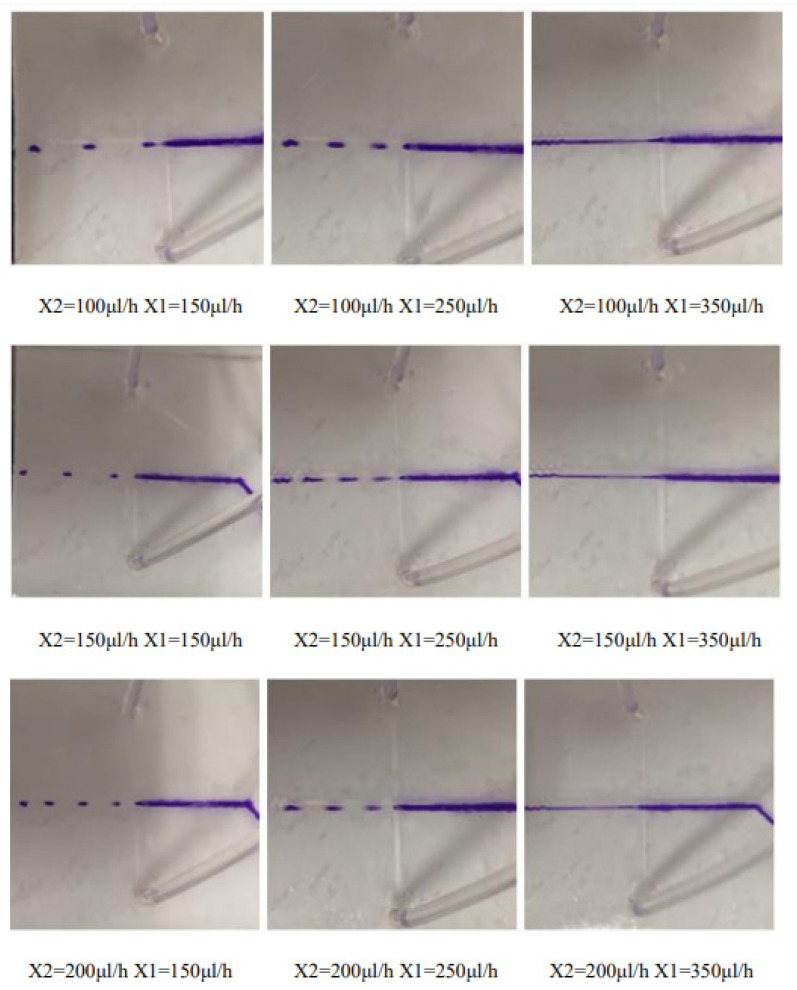
The experimental results of flow rate of the continuous phase being constant and the flow rate of the dispersed phase changing.

**Figure 13 micromachines-13-02050-f013:**
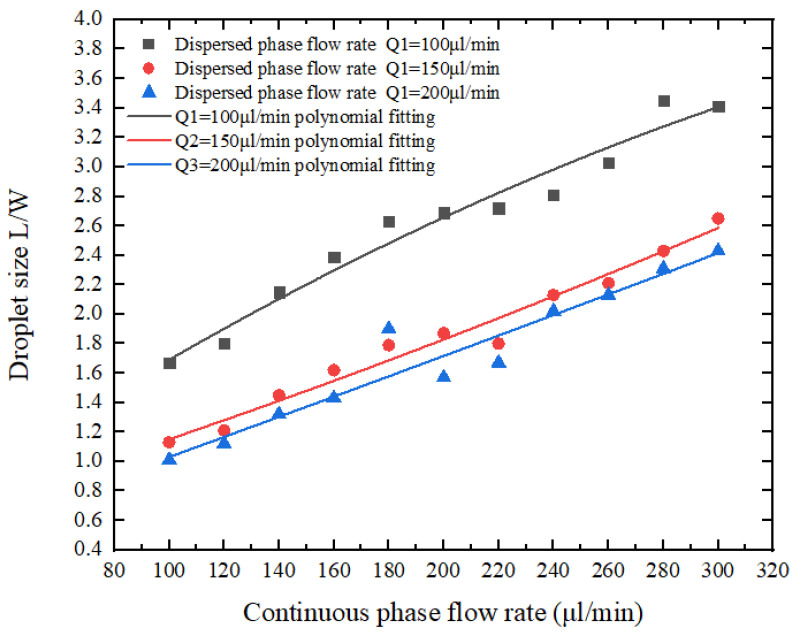
The relationship between the flow velocity of the dispersed phase and droplet size.

**Figure 14 micromachines-13-02050-f014:**
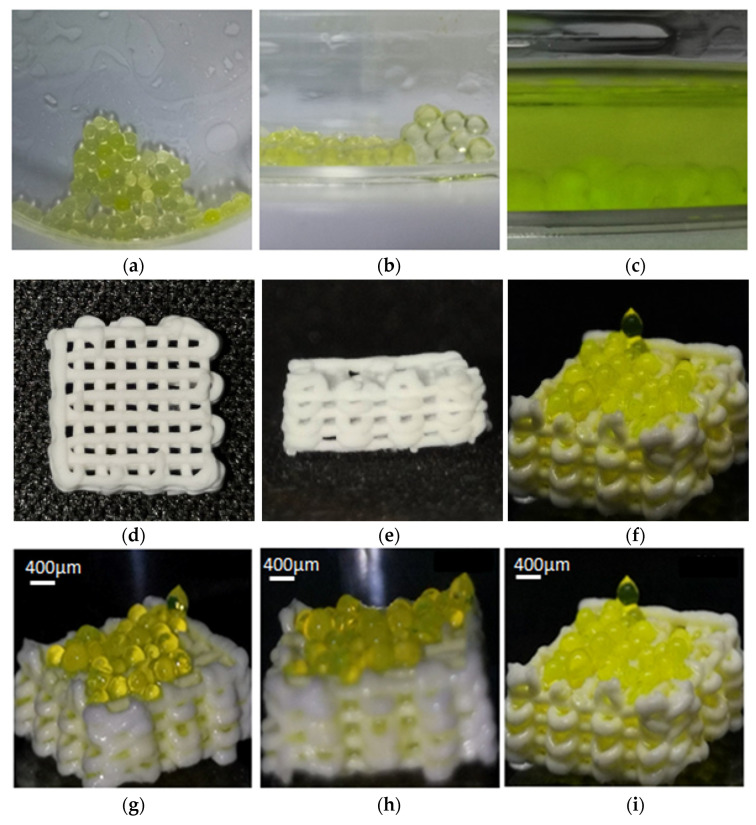
The state of cross-linked micro-droplets. (**a**,**b**) The morphology of microsphere droplets. (**c**) The shape of microsphere droplets in calcium chloride solution. (**d**,**e**) Biological scaffold. (**f**–**i**) The microsphere droplets are fixed on the biological scaffold superior.

**Figure 15 micromachines-13-02050-f015:**
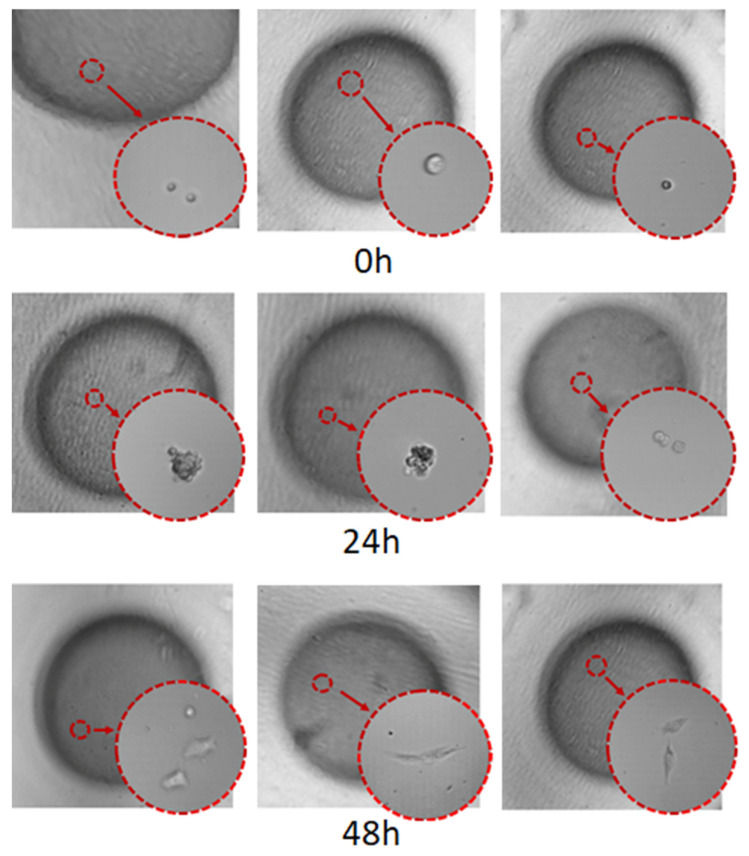
Cell culture in the droplet.

**Table 1 micromachines-13-02050-t001:** Basic parameters of the two fluids.

Properties	Fluid 1	Fluid 2
Viscosity (pa.s)	0.6	0.671
Density (g/mL)	1.5	0.877
Interfacial tension (n/m)	0.03	0.03
Contact angle	135°	135°

## Data Availability

Not applicable.

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
