# Peer review of "Rapid Customization and Manipulation Mechanism of Micro-Droplet Chip for 3D Cell Culture"

_micromachines, 2022, doi:10.3390/mi13122050_

Round 1

Reviewer 1 Report

This manuscript presents a rapid customization and manipulation mechanism of PDMS micro-droplet chip for 3D cell culture based on SLA light-curing 3D printing technology. The authors verified the performance of the full PDMS micro-droplet chip in biological culture. This manuscript can be improved based on the following comments:

1.-In the introduction section, the authors can add more information about research papers on cell culture using microfluidic chip method.

2.-Which are the advantages and limitations of the proposed microfluidic chip method compared to other methods reported in the technical literature?

3.-The resolution quality of figures 1, 2, 3, and 4 must be enhanced.

4.-The discussions of the results of figures 12 and 14 must be improved.

5.-The third section "experiments and results" is short. This section should add more information on the experiments and discussions on the main results.

6.-There are two Figures 15.

7.-The third section should consider more results of the proposed microfluidic chip method.

8.-Which are the challenges of the proposed microfluidic chip method?

Reviewer 2 Report

My general evaluation of the article titled “Rapid Customization and Manipulation Mechanism of Micro-droplet Chip for 3D Cell Culture” is as follows.

  It is a study in the field of “PDMS micro-droplet chip for 3D cell culture-fabricated 3d printer”. It is seen that the study was organized and written in accordance with its purpose. This study can be published in your journal as it is, but making the following corrections will strengthen the article.

1. The abstract should be revised. The technique-method can be stated briefly. The aim is not clearly stated.

2. In general, the English language of this article should be corrected. Professional help is recommended.

3. For the Materials-Methods section:

a- Why were the physical properties of the material used in this study not given in the form of a table?

b- Why were the geometric dimensions of the designed system not given?

c- How did you analyze Comsol without these features(materials-geometry)?

d- Can you show the dimensions in Figure 7-8-9?

4. Conclusions section should be developed. The superiority-difference between this article from other existing studies should be clearly stated.

5. Make use of recent studies in the References section.

Round 2

Reviewer 1 Report

The authors have improved the second version of their manuscript considering the reviewer's comments.